# Extracellular interactions and ligand degradation shape the nodal morphogen gradient

Yin Wang[1,2†], Xi Wang[3†], Thorsten Wohland[2,3*], Karuna Sampath[1*]

[1]Division of Biomedical Sciences, Warwick Medical School, University of Warwick, Coventry, United Kingdom; [2]Department of Biological Sciences, National University of Singapore, Singapore, Singapore; [3]Department of Chemistry, Centre for Bioimaging Sciences, National University of Singapore, Singapore, Singapore

**Abstract** The correct distribution and activity of secreted signaling proteins called morphogens is required for many developmental processes. Nodal morphogens play critical roles in embryonic axis formation in many organisms. Models proposed to generate the Nodal gradient include diffusivity, ligand processing, and a temporal activation window. But how the Nodal morphogen gradient forms in vivo remains unclear. Here, we have measured in vivo for the first time, the binding affinity of Nodal ligands to their major cell surface receptor, Acvr2b, and to the Nodal inhibitor, Lefty, by fluorescence cross-correlation spectroscopy. We examined the diffusion coefficient of Nodal ligands and Lefty inhibitors in live zebrafish embryos by fluorescence correlation spectroscopy. We also investigated the contribution of ligand degradation to the Nodal gradient. We show that ligand clearance via degradation shapes the Nodal gradient and correlates with its signaling range. By computational simulations of gradient formation, we demonstrate that diffusivity, extra-cellular interactions, and selective ligand destruction collectively shape the Nodal morphogen gradient.

**\*For correspondence:** twohland@nus.edu.sg (TW); K.Sampath@warwick.ac.uk (KS)

[†]These authors contributed equally to this work

**Competing interests:** The authors declare that no competing interests exist.

## Introduction

In many animals, development from a single cell to a complex multicellular organism requires the graded distribution and activity of diffusible proteins, which are known as morphogens. How morphogen gradients are formed is not fully understood. Studies in many organisms have suggested three major mechanisms to establish morphogen gradients: 1) diffusion, 2) transcytosis and 3) via cytonemes (*Green, 2002*; *Müller et al., 2013*; *Rogers and Schier, 2011*; *Wartlick et al., 2009*). For example, the gradient of fibroblast growth factors (FGFs) is established by diffusion and is regulated by extracellular heparan sulfate proteoglycans (HSPGs) (*Dowd et al., 1999*; *Duchesne et al., 2012*; *Makarenkova et al., 2009*; *Miura et al., 2009*; *Nowak et al., 2011*; *Yu et al., 2009*), whereas the gradient of *Drosophila* Decapentaplegic (Dpp) is established not only by diffusion, but also via transcytosis (*Dierick and Bejsovec, 1998*; *Kruse et al., 2004*) and cytonemes (*Hsiung et al., 2005*; *Ramirez-Weber and Kornberg, 1999*; *Roy et al., 2011*).

Nodal proteins, which belong to the TGF-β family of signaling proteins, play critical roles in vertebrate development (*Arnold and Robertson, 2009*; *Wakefield and Hill, 2013*). They serve as mesendoderm inducers in vertebrates, and are involved in many aspects of embryonic axis formation during development (*Kumari et al., 2013*; *Sampath and Robertson, 2016*). Nodal proteins are translated as precursors and function as dimers (*Massagué, 1990*). The Nodal precursors are cleaved by extracellular convertases, and convertase processing was found to be essential for Nodal activation in zebrafish and mouse embryonic tissues (*Beck et al., 2002*; *Le Good et al., 2005*). A

**eLife digest** Animals develop from a single fertilized egg cell into multicellular organisms. This process requires chemical signals called "morphogens" that instruct the cells how to behave during development. The morphogens move across cells and tissues to form gradients of the signal. Cells then respond in different ways depending on how much of the signal they receive. This, in turn, depends on several factors: first, how quickly or slowly the signal moves; second, how well the morphogen binds to responding cells and other molecules in its path; and third, how much signal is lost or destroyed during the movement.

Many researchers study morphogen gradients in the transparent zebrafish, since it grows quickly and it is easy to see developmental changes. However, until now it was not fully clear how the well-known morphogen called Nodal moves in live zebrafish as they develop.

Wang, Wang et al. have now investigated how well Nodal signals bind to the surface of cells that receive the signal and to a molecule called "Lefty", which is present in the same path and interferes with Nodal signals. Advanced techniques called fluorescence correlation and cross-correlation spectroscopy were used to measure Nodal signals at the level of single molecules in growing zebrafish. The experiments gave insights into how far Nodal signals move and remain active. The results showed that, in addition to Nodal diffusing and binding to receiving cells, one of the most important factors determining how far and quickly Nodal moves is its inactivation and destruction. Lastly, Wang, Wang et al. built computational models to test their observations from live zebrafish.

The current work was based on forcing zebrafish to produce molecules including Nodal at locations within the fish that normally do not make them. Therefore future experiments will aim to examine these molecules and their interactions when they are produced at their normal locations in the animal over time.

recent report found that FurinA convertase activity regulates long range signaling by the zebrafish left-right patterning Nodal, Southpaw (Spaw), but not other Nodal factors (*Tessadori et al., 2015*). Upon activation, Nodal proteins form complexes with type II and type I Activin receptors (Acvr1b; Acvr2a/b), which are serine/threonine kinases (*Reissmann et al., 2001*; *Yan et al., 2002*; *Yeo and Whitman, 2001*) and activate the Nodal pathway (*Jia et al., 2008*; *Kumar, 2000*; *Massagué et al., 2005*; *Whitman, 1998*). Nodal target genes include *nodal* itself and *lefty*, which encodes a feedback inhibitor of Nodal signaling (*Branford and Yost, 2002*; *Meno et al., 1999*). In zebrafish, of the three nodal homologs, *cyclops (cyc)*, *squint (sqt)* and *southpaw (spaw)*, *sqt* and *cyc* are expressed in an overlapping pattern in the gastrula margin where presumptive mesoderm and endoderm cells are located (*Erter et al., 1998*; *Feldman et al., 1998*; *Gritsman et al., 2000*; *Long et al., 2003*; *Rebagliati et al., 1998a*; *1998b*; *Sampath et al., 1998*; *van Boxtel et al., 2015*). However, Sqt and Cyc elicit differential responses in target cells: Sqt acts at long-range whereas Cyc only affects cells immediately adjacent to the source of the signal (*Chen and Schier, 2001*; *Jing et al., 2006*; *Müller et al., 2012*; *Tian et al., 2008*).

So far, there is no evidence for a requirement for transcytosis and cytonemes in distributing the Nodal factors and the Nodal morphogen gradient has been proposed to be established by simple diffusion (*Williams et al., 2004*). The diffusion coefficient of a molecule is a measure of its ability to move freely across a defined region. The free diffusion coefficient of the zebrafish Nodals has been suggested to be faster than their effective diffusion coefficient (*Müller et al., 2012*; *2013*), resulting in fast diffusion over short distances but slow diffusion over longer distances presumably by morphogen trapping at high affinity binding sites. These observations led to the hypothesis that Nodal diffusion is hindered either by cell surface interactions or by molecules in the extracellular matrix (*Müller et al., 2013*). How Nodal diffusion is hindered, and to what extent it shapes the Nodal gradient is unclear.

In contrast to the differential diffusion model, a recent study suggested that a temporal signal activation window created by microRNA-430 (miRNA-430) delays translation of the Nodal antagonist Lefty to determine the dimensions of Nodal signaling in the gastrula (*van Boxtel et al., 2015*). Repression by miRNA-430 likely plays a key role in regulation of Nodal signaling. However, miRNA-430 is not exclusive to *lefty1* but also targets *nodal/sqt* (*Choi et al., 2007*). Moreover, reporter protein

expression and ribosome-profiling data from zebrafish embryos indicate that Nodal/Sqt and Lefty1 are translated in a similar temporal window in the early gastrula (*Choi et al., 2007*; *Bazzini et al., 2012*; *Chew et al., 2013*). As such, it is unclear how the proposed temporal activation window might be converted into a spatial Nodal gradient.

Some studies have suggested that in addition to diffusion, the gradient of a morphogen is related to the rate of ligand clearance or stability (*Callejo et al., 2006*; *Chamberlain et al., 2008*; *Gregor et al., 2007*; *Kicheva et al., 2007*; *Wartlick et al., 2009*), and a role for stability and clearance of Nodals in vivo has been proposed (*Jing et al., 2006*; *Le Good et al., 2005*; *Tian and Meng, 2006*). Previously, we reported an atypical lysosome-targeting region located in the pro-domain of Cyc, which targets this Nodal protein for destruction, and regulates target gene induction (*Tian et al., 2008*). How the lysosome-targeting region regulates Nodal clearance and how it influences the Nodal morphogen gradient was not known.

In this study, we have examined the diffusion coefficient of Nodals in live zebrafish embryos by fluorescence correlation spectroscopy (FCS). FCS is a widely used single molecule sensitive technique that can quantitatively measure diffusion and concentrations in vivo by determining how fast particles diffuse through a fixed observation volume (*Shi et al., 2009c*; *Yu et al., 2009*). We estimated the affinity of Nodals to the type II receptor Acvr2b on the cell surface and to Lefty inhibitors in the extracellular space by single wavelength fluorescence cross-correlation spectroscopy (SW-FCCS). SW-FCCS uses a single laser to excite proteins labeled with spectrally different fluorophores within the observation volume (e.g., the confocal volume) (*Shi et al., 2009a*). By analyzing the correlated movement of various labeled proteins (ligands/receptor/inhibitor), we determined the fraction of the proteins that were free or bound, and calculated the dissociation constants in live zebrafish embryos. We also investigated the contribution of ligand stability in forming the Nodal gradient. By analyzing diffusion and binding in vivo and from computational simulations, we show that diffusivity alone is insufficient to generate the Nodal morphogen gradient. Our findings show that in order to generate and maintain a robust Nodal morphogen gradient, ligand clearance by degradation is balanced against the binding and release of Nodal ligands with the receptor and inhibitors.

## Results

### Nodal ligands demonstrate similar mobility profiles

To visualize Nodal ligands in vivo, we fused the enhanced green fluorescent protein (EGFP) with Sqt, Cyc, Sqt$^{Cyc2}$ and Cyc$^{\Delta2}$. The Cyc$^{\Delta2}$ mutant, which lacks a lysosomal targeting region in the Cyc pro-domain, shows significantly increased stability and signaling range over wild type Cyc protein (*Tian et al., 2008* and *Figure 1*). Sqt$^{Cyc2}$ chimeric protein harbors the atypical lysosome-targeting region from Cyc, and shows reduced stability and signaling range in comparison to Sqt (*Tian et al., 2008*). We tested the activity of the fusion proteins by comparing nodal target gene induction by the various fusion proteins to that of their untagged counterparts, and found similar activity (*Figure 1A–D* and *Figure 1—figure supplement 1*).

The diffusivity of extracellular signaling molecules can determine their distribution and activity range. To examine the diffusivity of the Nodal ligands, we determined the diffusion coefficients of Sqt-, Cyc-, Sqt$^{Cyc2}$- and Cyc$^{\Delta2}$-EGFP fusion proteins in vivo using FCS. EGFP-tagged Nodal fusion proteins were expressed from a localized source and FCS measurements were acquired in the extracellular space at various distances from the source cells (*Figure 1* and *Figure 1—source data 1*). All the Nodal-GFP fusions, including Sqt, Cyc, Sqt$^{Cyc2}$ and Cyc$^{\Delta2}$, as well as Lefty1 and Lefty2 fusions show very similar diffusion coefficients (*Figure 1F*). These results suggest that the free diffusivity alone is unlikely to differentiate the range and activity of the Nodal proteins.

### Sqt binds to Acvr2b receptor with higher affinity compared to Cyc

Cell surface receptors of extracellular signaling molecules can bind to the diffusible ligands, and thereby reduce their distribution range. In this scenario, the mobility of ligands that bind with higher affinity to the receptors will be more effectively retarded. To quantitate the binding affinity of Nodal ligands to the receptors, we determined the apparent dissociation constant ($K_d$) of Sqt and Cyc in vivo (*Foo et al., 2012b*) with the predominant Nodal receptor Acvr2b, using FCCS. To uncouple binding from signaling events within the cytoplasm, we fused the extracellular and trans-membrane

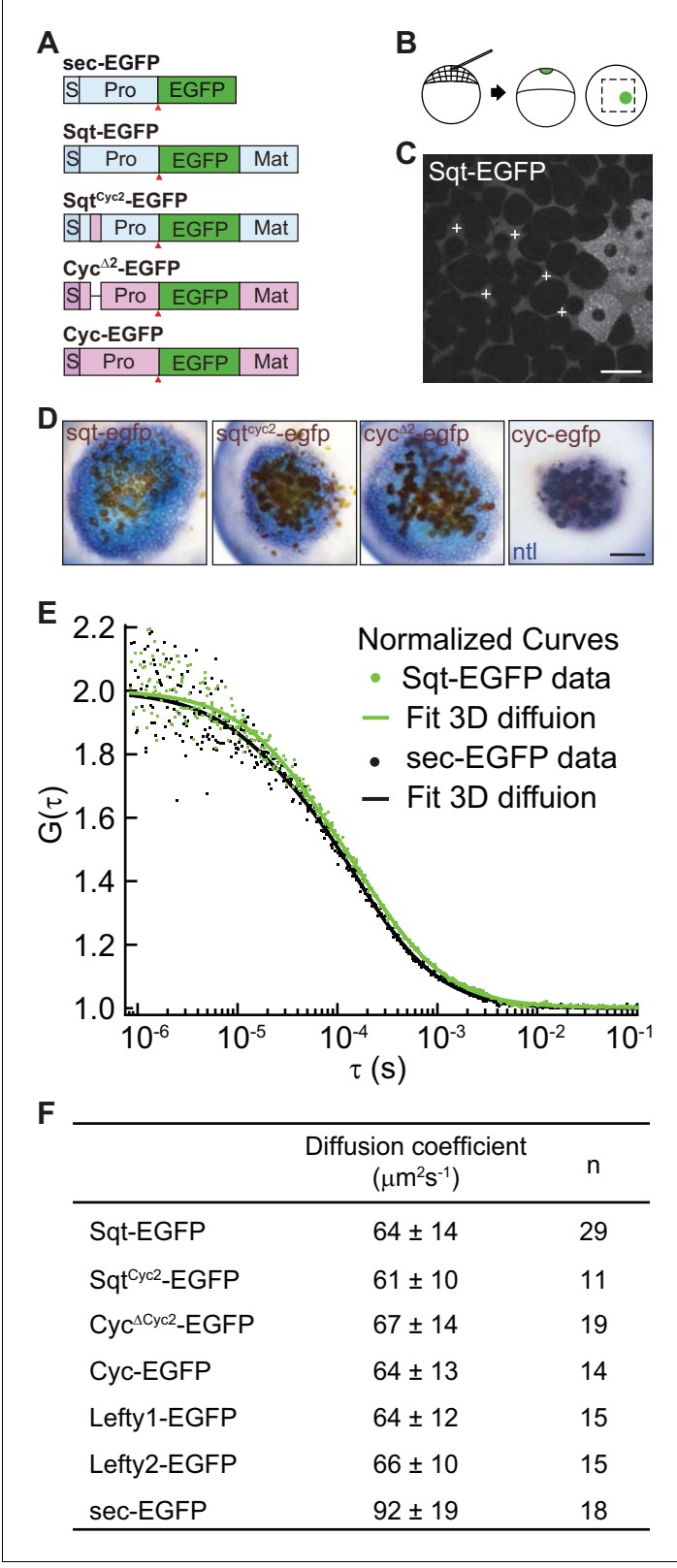

**Figure 1.** Activity range and diffusion of Nodal-GFP fusion proteins. (**A**) Constructs used for profiling fluorescent Nodal fusion proteins in embryos. S, signal peptide; Pro, pro-domain; Mat, mature-domain; sec-EGFP, secreted EGFP. Red arrow indicates convertase cleavage sites. (**B**) Injection procedure. (**C**) Confocal image of an injected embryo at 30% epiboly stage. White crosses mark the extracellular spots where the FCS measurements were
*Figure 1 continued on next page*

*Figure 1 continued*

taken. (D) Representative images of RNA in situ hybridization showing the activity range of Sqt, Cyc and mutant Nodals. Source cells are marked in brown and blue staining indicates expression of the Nodal target *ntl.* Scale bars, 50 µm. (E) Representative auto-correlation functions (dots) and fittings (line) of Sqt-EGFP (green) and sec-EGFP (black). (F) Table showing diffusion coefficients of the Nodal and Lefty fusion proteins as measured by FCS.

The following source data and figure supplement are available for figure 1:

**Source data 1.** Individual FCS measurements and diffusion coefficient values for EGFP-tagged Nodals and Leftys compared to control secreted EGFP.
**Figure supplement 1.** (Related to Main *Figure 1*).

domains of Acvr2b lacking the intracellular kinase domain, to a red fluorescent protein, mCherry (*Figure 2A–C*). Sqt-EGFP, Cyc-EGFP or control secreted eGFP (sec-EGFP) fusion proteins were expressed from a localized source in embryos that uniformly expressed Acvr2b-mCherry, and correlation curves were obtained to infer the $K_d$. Surprisingly, the $K_d$ of Sqt-Acvr2b is 65 ± 7 nM and the $K_d$ of Cyc-Acvr2b is 124 ± 12 nM (*Figure 2D–I*). This result suggests Sqt binds with Acvr2b with an approximately twofold higher affinity compared to Cyc.

## Sqt binds to Lefty2 inhibitor with higher affinity compared to Cyc

The Nodal antagonist Lefty prevents Nodal proteins from binding to their receptors and has the potential to influence the distribution of Nodal ligands. To test if binding to the inhibitor affects Nodal ligand distribution, we determined the affinity of Sqt and Cyc to Lefty2 in vivo by co-expressing Lefty-mCherry with Sqt-EGFP or Cyc-EGFP from a localized source and measuring the $K_d$ in the extracellular space of embryonic blastula cells at various distances from the source (*Figure 3A–C*). The $K_d$ of Sqt-Lefty2 is 29 ± 1.2 nM and Cyc-Lefty2 $K_d$ is 50 ± 3 nM (*Figure 3D–I*), indicating an approximately twofold higher affinity of Sqt-Lefty2 binding in comparison to Cyc-Lefty2 binding. The differential affinity of the Nodals for Lefty could fine-tune their activity range by removing freely diffusing Nodals from the signaling pool.

## The range of Nodal factors correlates with their stability

To visualize Nodal gradients in zebrafish embryos, we expressed Sqt, Cyc, Sqt$^{Cyc2}$ and Cyc$^{\Delta2}$-EGFP fusion proteins from a localized source (*Figure 4A–D*). Consistent with findings by Müller et al., Sqt-EGFP was found to reach the edges of the blastula with no more than 50–60% loss in intensity, whereas the intensity of Cyc-EGFP fusion protein falls steeply from the source (*Figure 4B* and *Figure 4—source data 1*). Interestingly, the gradient of the deletion mutant, Cyc$^{\Delta2}$-EGFP, which has a longer signaling range than Cyc-EGFP, was significantly shallower than that of Cyc, and the gradient of the Sqt$^{Cyc2}$-EGFP chimera (which has reduced signaling range compared to Sqt) was steeper than that of Sqt-EGFP (*Figure 4B* and *Figure 4—source data 1*).

To determine the relative stability of Sqt, Cyc, Cyc$^{\Delta2}$ and Sqt$^{Cyc2}$, we expressed FLAG-tagged versions of the proteins in HEK293T cells and examined the amount of secreted protein in the supernatant after various periods (*Figure 4C,D* and S2). By fitting the normalized band intensity acquired from immune-blots with an exponential decay model, we inferred the degradation rates ($0.166 \times 10^{-4}$ /s for Cyc-FLAG, $0.093 \times 10^{-4}$ /s for Cyc$^{\Delta2}$-FLAG $0.090 \times 10^{-4}$ /s for Sqt$^{Cyc2}$-FLAG and $0.003 \times 10^{-4}$ /s for Sqt-FLAG). The decay rate of these proteins shows a trend consistent with their gradient profile and their signaling range (*Figure 4A–D*, *Figure 4—source data 1*, *Figure 1* and S2). These results indicate a strong correlation between Nodal ligand stability and the gradient.

## Simulation of the Nodal gradient

To test the validity of our measurements, we performed simulations to model the Nodal gradient. Modeling of the Nodal morphogen gradient requires a range of different parameters, of which some have been measured in vivo and are available, and we have in this study determined binding affinities, and inferred concentrations (*Table 1*). We found the dissociation constants, $K_D$, of Sqt and Cyc to their major cell surface receptor Acvr2, to be ~60 and 120 nM, respectively. In addition, we

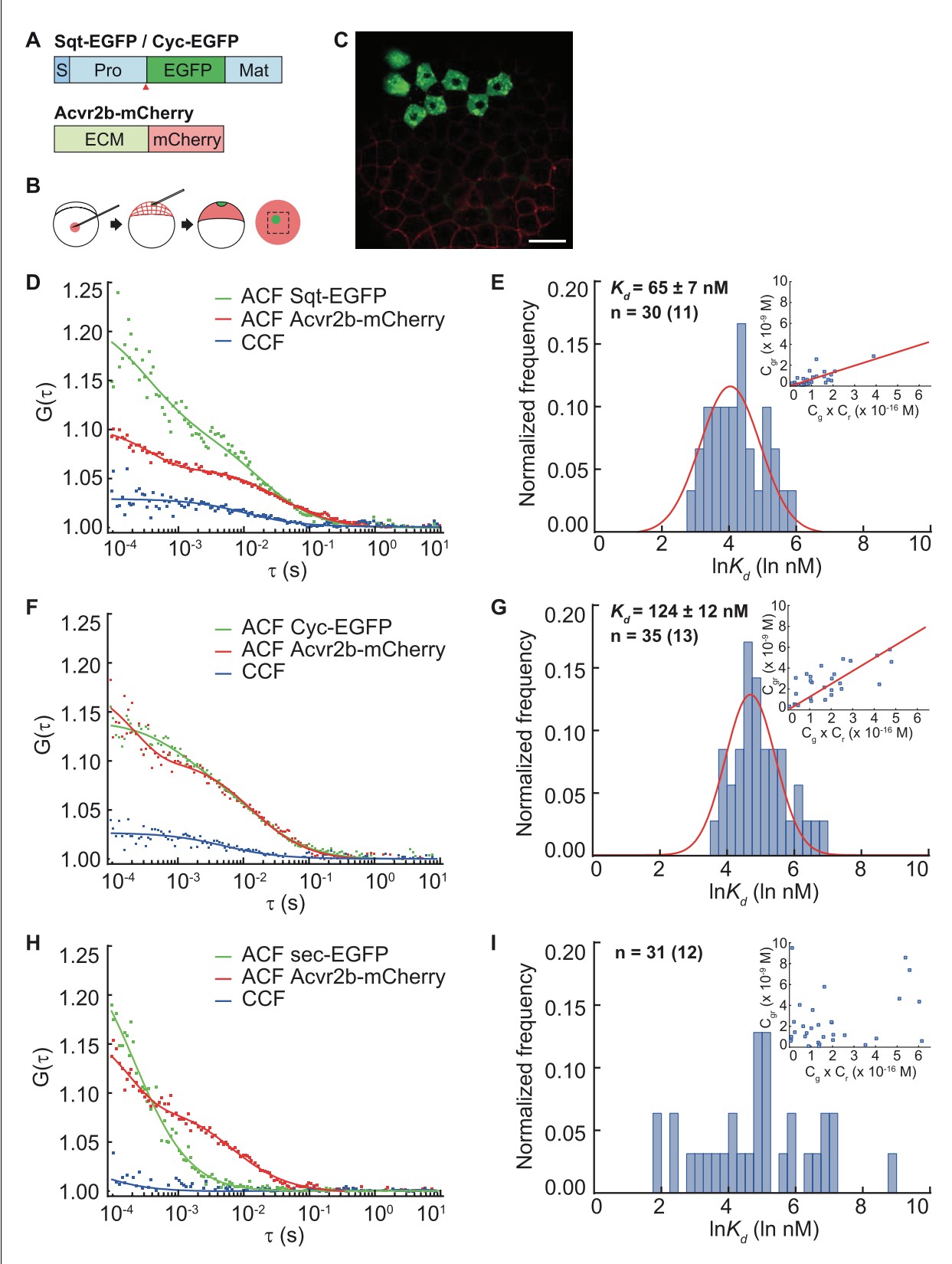

**Figure 2.** Sqt has higher affinity to the Acvr2b receptors than Cyc. (**A**) Sqt-/Cyc-/sec-EGFP and Acvr2b-mCherry constructs. S, signal peptide; Pro, pro-domain; Mat, mature-domain; ECM, extracellular and transmembrane domain. Red arrows indicate convertase cleavage sites. Sqt signal peptides and

Figure 2 continued

pro-domain were used in sec-EGFP constructs. (B) Injection procedure. (C) Representative image of an injected embryo at 30% epiboly stage showing the expression patterns of the fusion proteins. Scale bar represents 50 µm. (D,E,F) Representative auto-correlation (ACF) and cross-correlation functions (CCF) and fits. (G,H,I) Individual Ln($K_d$) frequency histogram and Gaussian fitting (red curve). Inset, concentration plot and linear regression (red line). X axis, concentration of bound protein ($C_{gr}$ (x10$^{-9}$ M)); Y axis, products of concentrations of free proteins ($C_g$ x $C_r$ (x10$^{-16}$ M)). n = number of data points (number of embryos).

determined the diffusion coefficients of the Nodals to be ~60 µm$^2$/s. From the amplitude of our FCS measurements, we estimated the concentration of the Nodal factors to be on the order of 10$^2$ nM. We determined that the degradation rate for Cyc is higher than that for Sqt, confirming previous work (*Tian et al., 2008*; *Jing et al., 2006*). Nonetheless, because our degradation rate values were estimated from cell culture, in our simulations we use the values of 0.0001/s and 0.0005/s for Sqt and Cyc, respectively, documented in or estimated from previous reports (*Jing et al., 2006*; *Müller et al., 2012*). In the simulations we produced particles with 0.07/s to 0.7/s in the simulation volume (corresponding to a production rate of 0.3–3 pM/s) to obtain sufficient number of particles for statistical analysis. The number of particles at equilibrium is given by the ratio of production over degradation rate, which was 700–7000 for a degradation rate of 0.0001/s and 140–1400 for 0.0005/s. Importantly, the production rate itself does not change the gradient shape and only the gradient amplitude is altered. Therefore, the gradient shape is determined by the degradation rate and diffusion.

First, we determined how the fact that particles have to transverse longer paths around obstacles (e.g., cells) during diffusion, renders diffusion apparently slower, and influences the effective diffusion coefficient (*Figure 5A,B* and *Videos 1*, *2*). This is referred to as tortuosity by *Müller et al. (2013)* and reduces diffusion maximally by a factor 2 (*Müller et al., 2013*). In agreement with this, we determined that for cells with 10 µm diameter and cell membrane-to cell membrane distance of 2 µm, we obtain a reduction of diffusion by a factor 1.84 (*Figure 5D*).

Secondly, we determined how binding affects the effective diffusion coefficient (*Figure 5C,E* and *Video 3*). For quantitative analysis, we simulated particles whose diffusion coefficient was recued from 60 to 30 µm$^2$/s due to tortuosity and assumed that a fraction of the particles is bound to binding sites that are homogeneously distributed. The effective diffusion coefficient is reduced more for higher affinities, i.e. when more particles are bound on average. For instance, when 90 or 99% of particles are bound, morphogen diffusion is reduced by a factor 10 or 100, respectively. The actual amount of bound ligand depends on the total concentration of ligand ($L_t$), receptor ($R_t$) and the $K_D$:

$$f_{bound} = \frac{K_d + L_t + R_t}{2L_t} - \sqrt{\frac{(K_d + L_t + R_t)^2}{4L_t^2} - \frac{R_t}{L_t}} \tag{1}$$

Thirdly, for the simulations we assume that Sqt and Cyc share the receptors, and a ligand concentration of 100 nM is used for both Nodal ligands. The receptor concentration is on the order of 10 µM or more (see next paragraph), and is much higher than that of the ligands. As such, the exact ligand amount does not change the outcome significantly for ligand concentration changes within a factor of ~5. The differences in the gradient length are therefore, a result of diffusion, differential binding of Sqt and Cyc, as well as different degradation rates.

Next, we determined the effective diffusion coefficient that results in a Sqt gradient length consistent with our measured values of about 30 µm. The gradient length is described by a model previously used for Fgf8 diffusion in zebrafish embryos (*Müller et al., 2013*; *Yu et al., 2009*):

$$\lambda = \sqrt{\frac{D/k}{R/K_d + 1}} \tag{2}$$

where λ is the gradient length, $D$ is the free diffusion coefficient, $k$ is the clearance rate which is assumed constant, $R$ is the concentration of the receptor, and $K_d$ is the equilibrium dissociation constant, respectively (*Table 1*). Using a gradient length of about 30 µm and the other values as given in *Table 1*, we estimate the effective diffusion coefficient to be on the order of 0.045 µm$^2$/s. With a $K_d$ of 60 nM for Sqt and an $L_t$ of 100 nM, this requires a bound fraction $f_{bound}$ of 99.85% and a

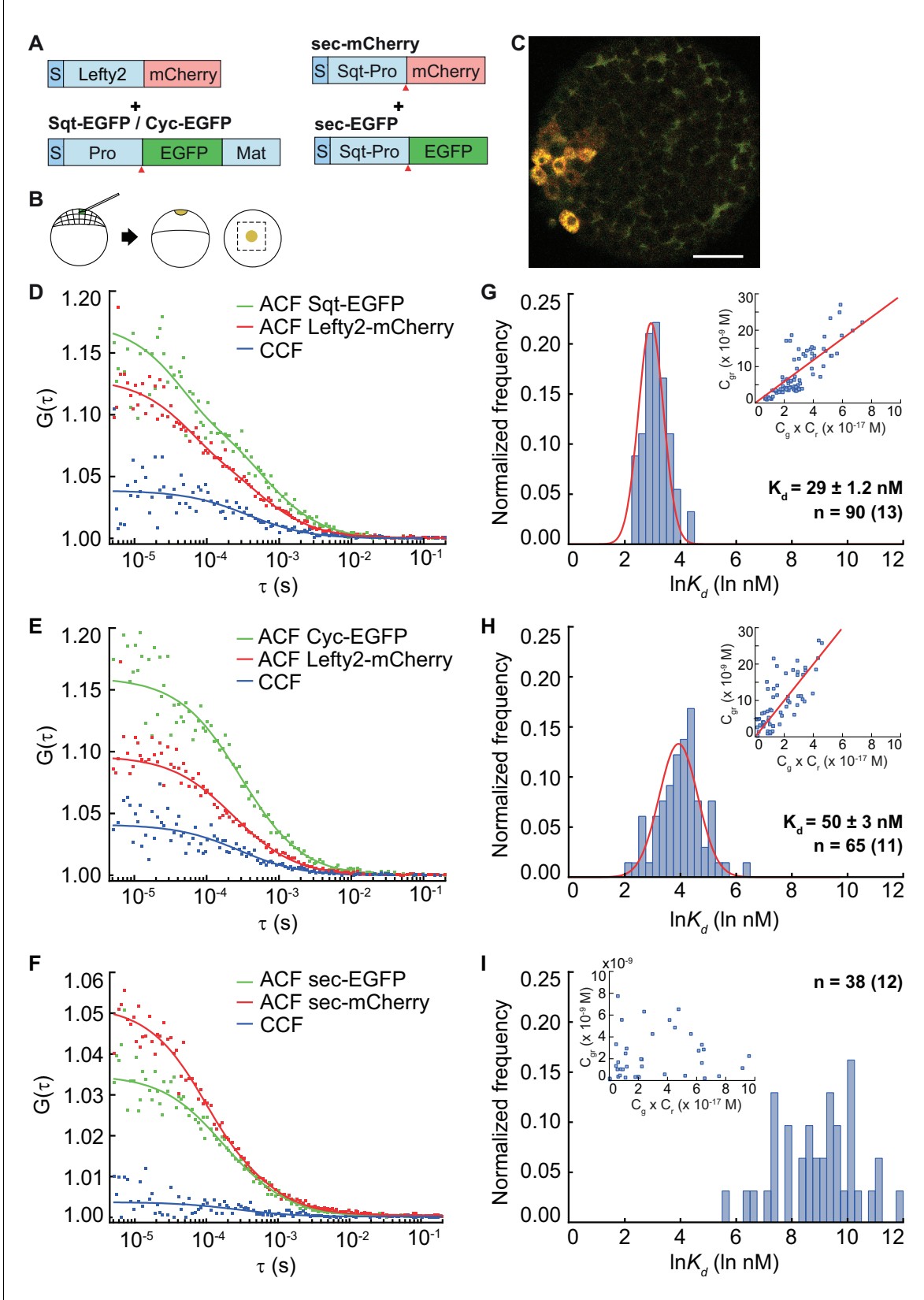

**Figure 3.** FCCS measurements reveal that Lefty has higher affinity to Sqt compared to Cyc. (**A**) Constructs used for injection. S, signal peptide; Pro, pro-domain; Mat, mature-domain. Red arrows indicate the convertase cleavage sites. (**B**) Injection procedure. (**C**) Confocal image of an injected embryo

*Figure 3 continued on next page*

*Figure 3 continued*

at 30% epiboly showing the expression patterns of the fusion proteins. Scale bar represents 50 μm. (D, E, F) Representative auto- and cross-correlation functions (ACF; CCF) and fittings. (G, H, I) Individual Ln($K_d$) frequency histogram and Gaussian fits (red curve). Inset, concentration plot and linear regression (red line). X axis, concentration of bound protein ($C_{gr}$(x10$^{-9}$ M)); Y axis, products of concentrations of free proteins ($C_g$ x $C_r$(x10$^{-17}$ M)). n = number of data points (i.e., number of embryos).

receptor concentration of ∼40 μM. The estimation of the receptor number of 40 μM is based on the value of $D_{eff}$ required to establish the gradient of appropriate dimensions for Sqt. At this time it is not clear whether this concentration comprises only membrane receptors or whether additional binding sites (e.g., in interstitial spaces between cells) contribute to it as shown for Fgf8 (*Yu et al., 2009*). Any corrections in binding affinities for the receptor or different affinities for additional binding sites would alter the required concentration. Importantly, this number is dependent on the clearance/degradation rates that we have based upon previous reports, and which could change with more precise *in vivo* measurements. Based on the above, we assume that 40 μM is the upper limit for the receptor concentration. At this receptor concentration, Cyc, with a $K_d$ of 120 nM will have a bound fraction of 99.7% and an effective diffusion coefficient of 0.09 μm$^2$/s.

Finally, we used these effective diffusion coefficients and the degradation rates of 0.0001/s and 0.0005/s for Sqt and Cyc, respectively, to simulate gradient formation (see simulation parameters in *Table 1*). Using these parameters, Sqt produced a gradient of ∼30 μm length and Cyc gradient was ∼19 μm (*Figure 5F*). These values are consistent with actual in vivo measurements (*Figure 4B* and *Figure 4—source data 1*), and support our hypothesis.

## Discussion

In our study, we found that the diffusion coefficients of free Nodal and Lefty proteins measured by FCS are very similar (∼60 μm$^2$/s), consistent with their similar apparent molecular weight and with previous reports (*Müller et al., 2013*). However, their mobility is too great to account for the sharp gradients observed in developing embryos. The diffusion coefficient values determined by FCS are ∼3 to 85 times higher than the effective diffusion coefficients reported using FRAP (18.9 ± 3.0 μm$^2$/s for Lefty2, 3.2 ± 0.5 μm$^2$/s for Sqt and 0.7 ± 0.2 μm$^2$/s for Cyc) (*Müller et al., 2013*). FCS and FRAP produce different readouts because they measure diffusion in different contexts, time windows and scales. FCS determines diffusion within a small volume (<1 μm$^3$) and on a short timescale (<1 s), whereas FRAP measures net diffusion over a large area (>1,000 μm$^3$) over a longer time period (>>10 s). FCS has been very useful to determine local diffusion, to infer the concentration of molecules within a defined confocal volume, and to determine the affinity of molecular interactions within the defined confocal volume. FRAP has been very useful for examining large-scale movement of molecules in tissues. The diffusion coefficients determined by the two techniques are known to vary dramatically. For example, in *Drosophila* imaginal discs, the measured diffusion coefficient of Dpp-GFP is 10 ± 1 μm$^2$/s from FCS measurements (*Zhou et al., 2012*), and 0.1 ± 0.05 μm$^2$/s by FRAP (*Kicheva et al., 2007*). This difference is thought to arise from other factors (e.g., by degradation of the molecules during diffusion). Taken together, these findings strongly suggest there must be other molecules and mechanisms in the embryo that refine and shape the Nodal morphogen gradient.

The diffusional movement of morphogens can also be altered by transient binding to other molecules such as receptors or to components of the extracellular matrix (*Baeg et al., 2004*; *Belenkaya et al., 2004*; *Han et al., 2004*; *Lander et al., 2002*; *Wang et al., 2008*; *Yu et al., 2009*), so that one possible mechanism to shape the gradient is transient binding of Nodal proteins to immobilized diffusion regulators, as found for the fibroblast growth factor Fgf8 (*Yu et al., 2009*). To explain the long range distribution of Sqt compared to Cyc, it was also proposed that Sqt might have a lower binding affinity to its receptors, (*Müller et al., 2012*). However, we find that Sqt in fact binds in vivo to Acvr2b with a higher affinity than the short-range Nodal, Cyc. Sqt also binds to Lefty with higher affinity, raising the possibility that Lefty-binding might alter Sqt activity. Another potential mechanism for gradient formation is rapid clearance of molecules during diffusion, as observed for Dpp in *Drosophila* (*Kicheva et al., 2007*). Our measurements and calculations strongly support this mechanism.

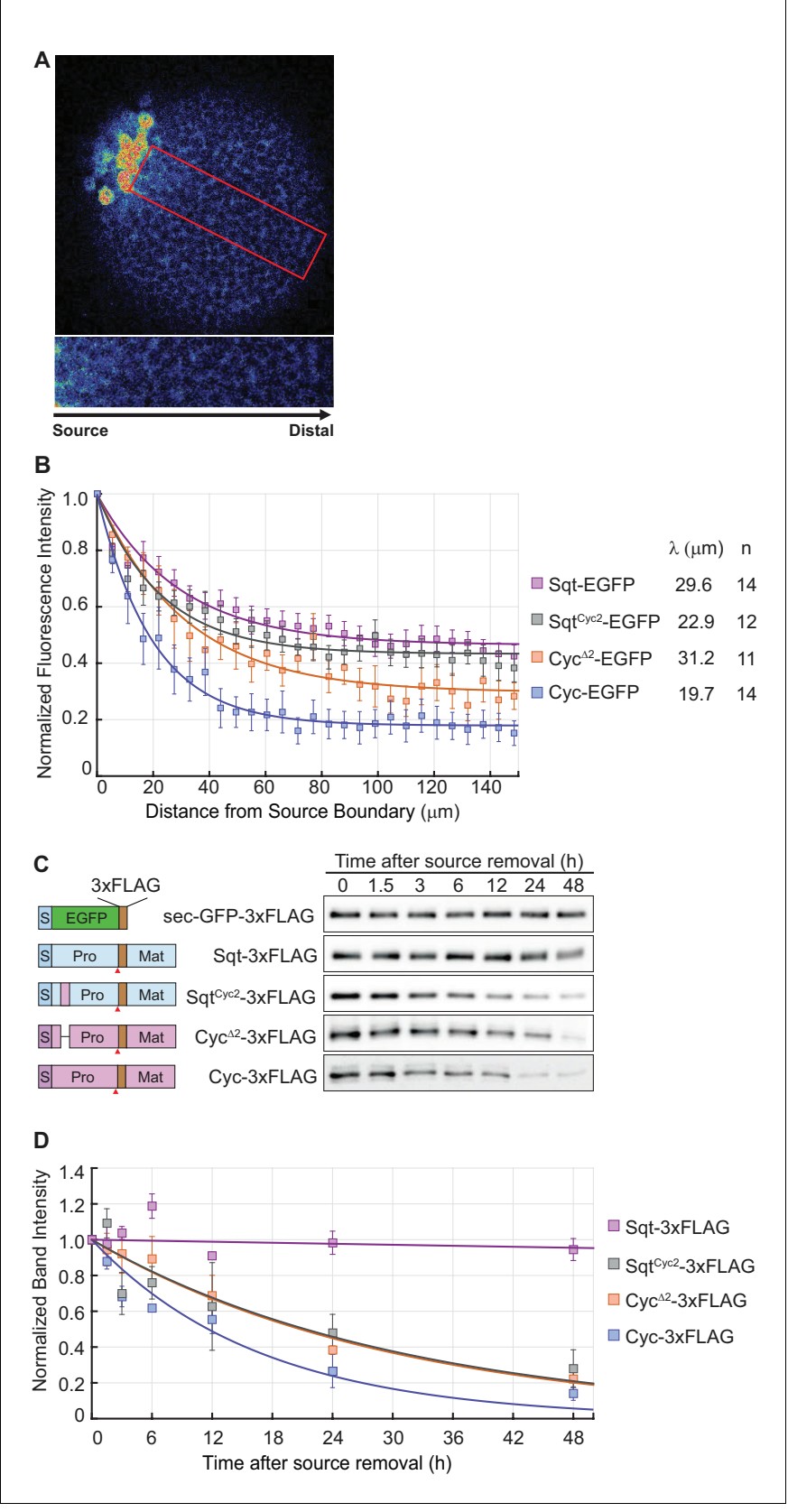

**Figure 4.** The distribution of Nodal proteins correlates with clearance. (A) Upper, representative image and region of interest (red rectangle) for measuring distribution; lower, inset showing magnified region of interest. (B) *Figure 4 continued on next page*

*Figure 4 continued*

Normalized distribution profiles and fitting. Error bars indicate standard error of mean (s.e.m). (**C**) Representative western blots of Nodal proteins harvested from HEK293T cell culture medium at different time points after removal of the source. The Nodal proteins were immuno-precipitated with anti-FLAG antibody and detected by western blot with the same antibody. Schematics on the left show the position of the FLAG epitope tags in each construct. (**D**) The profile of Nodal protein levels over time after source removal. The data points were fitted with an exponential decay model. Error bars indicate s.e.m.

The following source data and figure supplement are available for figure 4:

**Source data 1.** Gradient data for tagged wild type and mutant Nodals.

**Figure supplement 1.** (Related to Main *Figure 4*).

Previous studies have suggested that the Nodal gradient might be influenced by its stability: Le Good *et al.* showed that increasing the stability of mouse Nodal protein increases its range of activity (*Le Good et al., 2005*); Tian *et al.* found a lysozyme targeting signal in Cyc that accelerates its degradation and reduces the signaling activity of chimeric Sqt$^{Cyc2}$ protein (*Tian et al., 2008*); Jing *et al.* determined the half-life of Sqt and Cyc to be ~8 hr (~480 min) and ~2 hr (~120 min), respectively, which somewhat correlates with the difference in target induction by the two proteins (*Jing et al., 2006*). However, the difference in clearance rates of Sqt, Cyc, Lefty1 and Lefty2 determined by photo-conversion assays is not pronounced enough to explain their very different decay lengths (*Müller et al., 2012*). Interestingly, Müller et al. found that their fluorescent Cyc fusion protein was expressed at very low levels in the extracellular matrix, but exhibited an unusual punctate distribution close to the plasma membrane and in the cytosol, whereas their Sqt fusion showed a strong, uniform and mainly intracellular distribution. The punctate distribution of Cyc suggests that Cyc might undergo a much faster and/or sustained endocytosis process compared to Sqt. This supports our finding that cells selectively destroy Nodal ligands by recognizing the lysosome-targeting signal, since the ligands have to be internalized.

Simulations of the Nodal gradient show that Sqt generates a gradient of 30 μm and Cyc 19.1 μm, consistent with our measurements of 29.5 ± 5 μm for Sqt and 19.7 ± 2 μm for Cyc (from *Figure 4B*), as well as the estimated signaling range of these proteins (*Chen and Schier 2001*; *Tian et al., 2008*). In the simulations, 80% and 95% of steady state levels for Cyc is achieved at 0.7 and 1.25 hr, respectively, which is consistent with the timing of mesoderm induction in the gastrula. By the same predictions, Sqt reaches the 80% and 95% levels at ~4 hr and 7 hr, respectively, which is longer than expected. However, these simulations have not taken into consideration cell divisions or binding to other factors that could influence the gradient. Despite some differences in absolute values, the overall agreement between our experimental results, theory, and simulation supports our conclusion that the Nodal gradient is dependent upon diffusion, binding, and degradation of the morphogen.

An important point to note from *Equation (2)* is that the clearance rate $k$ and the receptor concentration $R$ are inversely related to each other. Thus, a higher clearance rate would predict a lower

**Table 1.** Simulation parameters

| Parameter | Sqt | Cyc | Reference |
|---|---|---|---|
| Kd | 60 nM | 120 nM | This work |
| Degradation rate | 0.0001/s | 0.0005/s | Estimated from *Jing et al., 2006*; *Müller et al., 2012* |
| Ligand concentration | ~100 nM | ~100 nM | Estimated from this work |
| Receptor concentration | 40 μM | 40 μM | Estimated from this work |
| $D_{free}$ [μm2/s] | 60 | 60 | This work |
| $D_{tortuosity}$ [μm2/s] | 30 | 30 | This work and *Müller et al., 2013* |
| $D_{eff}$ [μm2/s] | 0.045 | 0.09 | This work |

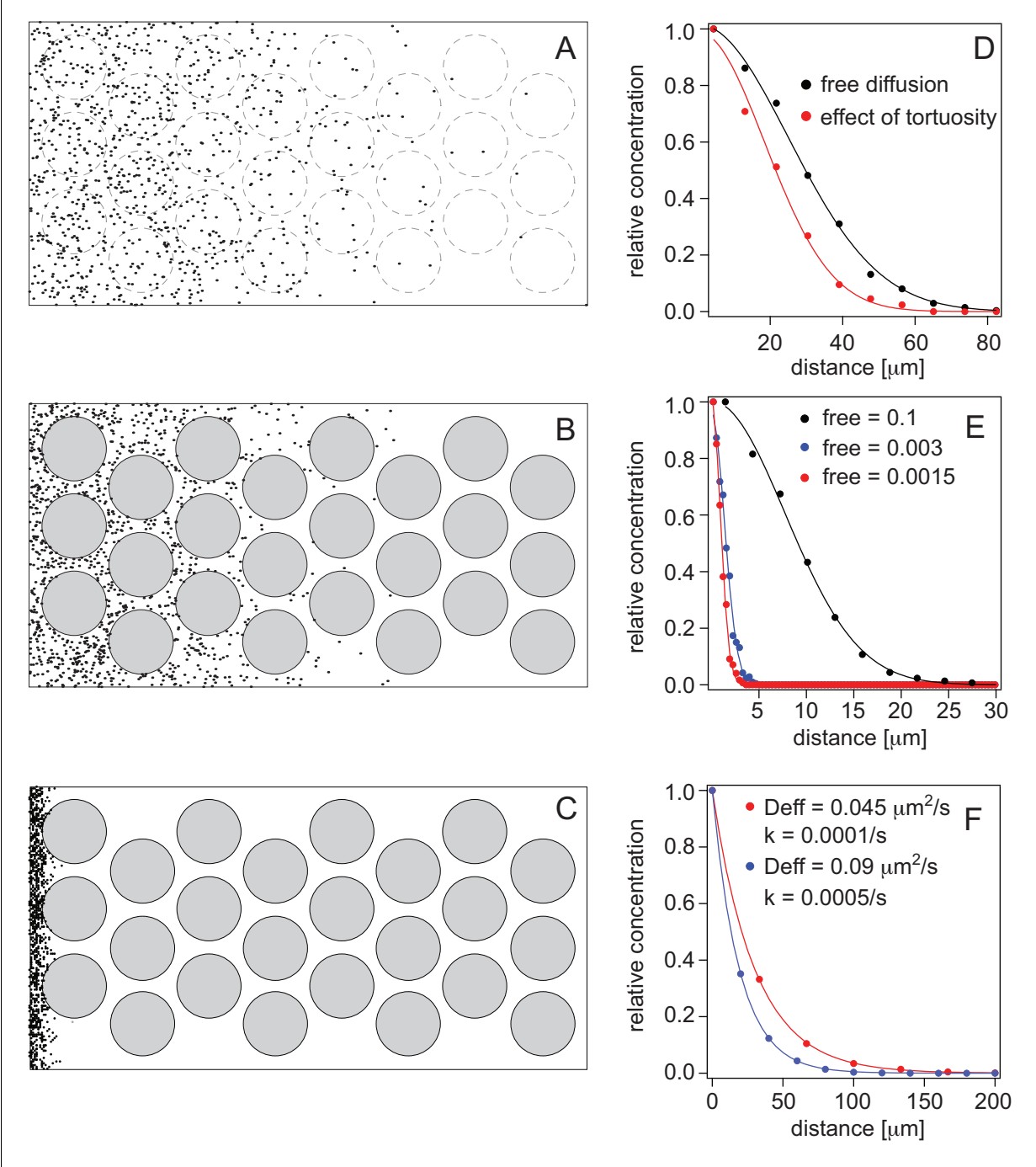

**Figure 5.** Simulations of morphogen diffusion. (**A**) Free diffusion with a diffusion coefficient $D = 60 \ \mu m^2/s$. (Dashed circles indicate positions of cells in later simulations but not taken account of in this case). (**B**) Diffusion in the presence of cells. (**C**) Diffusion in the presence of cells and binding with an average number of free particles of 0.003, i.e. 99.7% of all particles are bound on average. Simulations were done in a 3D space as described in the text and the diffusion coefficient was $D = 60 \ \mu m^2/s$. (**D**) Comparison of the spread of the particles as a function of the distance from the source (the left border in panels **A**–**C**). The concentration curves were fit with a bell curve that describes the diffusion of particles from the source. For free diffusion (**A**) we recover a diffusion coefficient of $D = 63.4 \ \mu m^2/s$ close to the input value, and in the presence of cells (**B**) this reduces to an effective diffusion coefficient of $D_{eff} = 33.8 \ \mu m^2/s$, demonstrating the effect of tortuosity. (**E**) Simulations of diffusion in the presence of cells, and with different amounts of binding. The simulated diffusion coefficient was $D = 60 \ \mu m^2/s$. The concentration curves were fit with *Equation 8*. The recovered effective diffusion coefficients for a fraction of free particles of 0.1, 0.003, and 0.0015 were $D_{eff} = 2.99 \ \mu m^2/s$, $D_{eff} = 0.09 \ \mu m^2/s$, and $D_{eff} = 0.042 \ \mu m^2/s$, respectively, demonstrating the effect of binding on the effective diffusion coefficient. (**F**) Gradient formation using the effective diffusion coefficients determined from graph **E** and degradation rates of 0.0001/s and 0.0005/s, respectively. The blue curve represents Cyc, the red curve Sqt. Although Sqt has higher

*Figure 5 continued on next page*

*Figure 5 continued*

binding affinity and consequently a lower free mobile fraction, its lower degradation rate ensures that Sqt has a less steep gradient. The data was fit with an exponential function yielding gradients of 19 μm for Cyc and and 30 μm for Sqt, respectively.

receptor concentration and vice versa. It will be interesting to determine these values with higher accuracy in live zebrafish embryos. In addition, some aspects of the system have not been taken into account in our simulations. In particular, we found that Lefty binds the Nodals with high affinity. This may not influence gradient formation as the Sqt/Lefty and Cyc/Lefty complexes likely diffuse very similarly to Sqt and Cyc given that their size difference is within a factor of two. However, Lefty will influence Sqt and Cyc signaling when in complex, even if this is not directly evident in the gradient of fluorescent molecules. We also have not considered how the Nodal co-receptor (*Yan et al., 1999*) influences gradient formation. It will be interesting to determine how Oep/Cripto co-receptors and Lefty shape the active signaling gradient. The extracellular matrix (ECM) has been shown to play a key role in regulating diffusion of FGFs, presumably via interactions with heparan sulphate proteoglycans (HSPGs) (*Makarenkova et al., 2009*; *Yu et al., 2009*). It is not known if the ECM or HSPGs play a role in modulating the Nodal morphogen gradient although sulfated proteoglycans have been proposed to provide directional cues for left-asymmetric Nodal in *Xenopus* (*Marjoram and Wright., 2011*).

In conclusion, we find that in addition to hindered diffusion via binding to the receptors and inhibitors, the differential stability of Nodal ligands play key roles in shaping the Nodal gradient and activity range. Our experimental findings together with theoretical and computational simulations show that diffusion, extracellular interactions i.e., Nodal-receptor binding, Nodal-Lefty inhibitor binding, and selective ligand destruction collectively shape and refine the Nodal morphogen gradient.

## Materials and methods

### Generation of constructs

All the constructs were PCR amplified and cloned into pCS2+ vector with Kozak sequence gccacc immediate 5' of the start codon. For Cyc and Cyc$^{\Delta2}$ fusions, EGFP or 3xFLAG (DYKDHDGDYKDHD-I-DYKDDDDK) tag was inserted 4 amino acids after the cleavage site (RRGRR). For Sqt and Sqt$^{Cyc2}$ fusions, EGFP or 3xFLAG tag was inserted 1 amino acid after the cleavage site (RRHRR). For Lefty1 and Lefty2 fusions, EGFP or 3xFLAG tag was fused to the C-terminus of the protein as previously described (*Müller et al., 2012*). For Acvr2b fusion, mCherry was fused to the C-terminus of Acvr2b (1–188 aa). For generating the sec-EGFP construct, the EGFP tag was fused to the C-terminus of 4 amino acids after the cleavage site (RRGRR) of Sqt. For sec-EGFP-3xFLAG construct, 3xFLAG tag was fused to the C-terminus of Sec-EGFP (*Yu et al., 2009*).

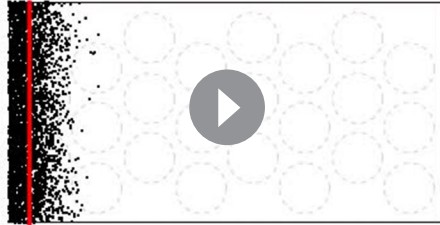

**Video 1.** Simulation of particles moving and freely diffusing from the source. Dashed circles indicate positions of cells in later simulations (included here for illustration only, but have no influence on the simulation).

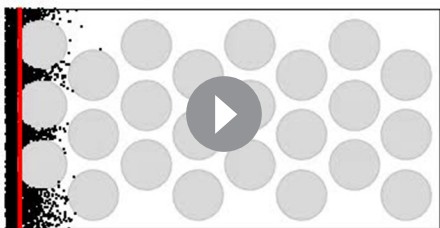

**Video 2.** Particle movement is hindered by cells (black circles) around which they move.

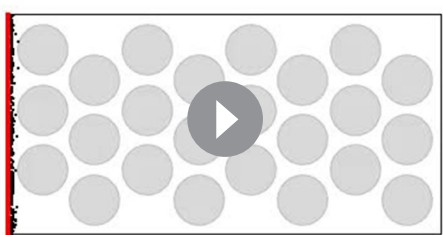

**Video 3.** Particle movement is hindered further by binding.

## Zebrafish strains

Wild-type (AB) fish were maintained at 28.5°C and embryos were obtained from natural matings according to standard procedures and in accordance with institutional animal care regulations.

## Capped RNA synthesis

The plasmids were linearized with NotI restriction endonuclease (NEB) and transcribed using the mMessage mMachine SP6 Kit (Ambion) to produce capped RNA. Synthetic RNA was purified with P-30 Bio-Spin columns (Bio-Rad, Hercules, CA) followed by phenol/chloroform extraction and ethanol precipitation, and RNA concentration was quantified by Nanodrop (Thermo Fisher Scientific, Waltham, MA) and estimation of agarose gel electrophoresis bands.

## Microinjection and sample preparation

To test overall inductivity of the various Nodal fusions, 5 ng of the RNA was mixed with 0.25% phenol red (Sigma, Aldrich, St Louis, MO) and injected into the yolk of 1-cell stage AB wild type embryos. To test the signaling range of the Nodal fusions, single cells of de-chorionated 128-cell stage embryos were injected with 2.5 pg RNA and 0.25% 10kDa biotin-Dextran (Thermo Fisher Scientific). Un-injected embryos from the same batch were used as controls and as reference for staging the embryos. Embryos were fixed at the 50% epiboly stage in 4% paraformaldehyde (PFA) in PBS at 4°C for at least 24 hr. The embryos were subjected to in situ hybridization to detect *gsc* and *ntl* expression levels as previously described (*Le Good et al., 2005*; *Müller et al., 2012*; *Tian et al., 2008*).

To generate clones of cells expressing Nodal or Lefty, Sqt, Cyc, Lefty1 or Lefty2 -EGFP RNA (2.5 pg) was injected into single cells of 32–128-cell stage embryos. To determine the dissociation constant of Nodal and Acvr2b, 50 pg RNA encoding Acvr2b-mCherry was injected into 1-cell stage embryos prior to clone generation. The embryos were mounted on glass-bottom dishes (World Precision Instruments) in 0.75% low melting temperature agarose (in 30% Danieau's solution) at the 30% epiboly stage for confocal imaging and FCS/FCCS measurements.

## Protein expression and detection

HEK293T cells were transfected with plasmid DNA encoding 3xFLAG tagged proteins using FuGene HD (Promega, Madison, WI) transfection reagent. The medium with transfection reagent was removed and replaced with fresh Opti-MEM medium (Life Technologies, Carlsbad, CA) 24 hr after addition of transfection reagent. Cell culture supernatants were collected and flash frozen in liquid nitrogen 24 hr after the removal of the transfection reagent. Small aliquots of frozen supernatants were immunoprecipitated with anti-FLAG M2 antibody (Sigma) and protein G dynabeads (Life Technologies), and eluted with 3xFLAG peptide (Sigma). The samples were immunoblotted with the same antibody and the signals were detected with a Syngene PXi gel imaging system. The band intensity was quantified using ImageJ. To determine the clearance rate of the proteins, the remaining supernatant was diluted to the same concentration as supernatants from non-transfected cells, mixed with sec-EGFP-3xFLAG supernatant for input control, added to dishes with non-transfected cells and collected at different time points. Proteinase inhibitor cocktail (Roche, Switzerland) was added immediately after the supernatants were collected and flash frozen in liquid nitrogen. The supernatants were enriched and detected as described above. The intensity of individual protein bands was normalized against EGFP to correct for differences in sample volume and immunoprecipitation, and normalized to time 0 for relative changes.

## FCS/FCCS instrumentation

A custom-built single wavelength fluorescence cross-correlation spectroscopy (SW-FCCS) system was used for the FCS and FCCS measurements as described (*Shi et al., 2009a*; *2009b*).

## Measurement of diffusion coefficients and binding affinity

We obtained the correlation curve of the various fusion proteins by focusing the detection volume on the cell membrane at various distances from the source (*Figure 2A–C*). The correlation curves were analyzed and fitted with a bimolecular binding model to calculate the $K_d$. We co-expressed Lefty-mCherry with Sqt-EGFP or Cyc-EGFP from a localized source and measured the $K_d$ in the extracellular space of blastula cells at various distances from the source.

## FCS/FCCS data processing

The experimental raw auto-correlation data was fitted with defined correlation function models.

In FCS, a one-component 3D diffusion model with triplet state was used for free diffusing molecules:

$$G_{3D,1C,1trip}(\tau) = \frac{1}{N}\left[1+\left(\frac{F_{trip}}{1-F_{trip}}\right)e^{-\tau/\tau_{trip}}\right]\left(1+\frac{\tau}{\tau_d}\right)^{-1}\left[1+\left(\frac{\omega_0}{z_0}\right)^2\frac{\tau}{\tau_d}\right]^{-1/2}+G_\infty, \qquad (3)$$

where $N$ is the number of particles in the confocal volume; $F_{trip}$ is the fraction of the particles that have entered the triplet state; $\tau_{trip}$ is the triplet state relaxation time; $\tau_d$ is the average time required for one particle to diffuse through the confocal volume, $\omega_0$ and $z_0$ are the radial and axial distances where the excitation intensity reaches $1/e^2$ of its value from the center of the confocal volume; and $G_\infty$ is the convergence value of the ACF for long times.

In FCCS, a one-component 2D diffusion model and a two-component 3D model were used for the membrane anchored receptors and Nodal ligands, respectively:

$$G_{2D,1C,1trip}(\tau) = \frac{1}{N}\left[1+\left(\frac{F_{trip}}{1-F_{trip}}\right)e^{-\tau/\tau_{trip}}\right]\left(1+\frac{\tau}{\tau_d}\right)^{-1}+G_\infty \qquad (4)$$

$$G_{3D,2C,1trip}(\tau) = \frac{1}{N}\left[1+\left(\frac{F_{trip}}{1-F_{trip}}\right)e^{-\tau/\tau_{trip}}\right]\left\{\sum_i F_i\left(1+\frac{\tau}{\tau_{di}}\right)^{-1}\left[1+\left(\frac{\omega_0}{z_0}\right)^2\frac{\tau}{\tau_{di}}\right]^{-1/2}\right\}+G_\infty \qquad (5)$$

where $\tau_{di}$ and $F_i$ are the diffusion time and the amplitude of the $i^{th}$ component. The cross-correlation data was fitted by a one-component 2D model:

$$G_{3D,1C,1trip}(\tau) = \frac{1}{N}\left(1+\frac{\tau}{\tau_d}\right)^{-1}+G_\infty \qquad (6)$$

Data was fit with the Levenberg-Marquardt algorithm using the described models in Igor Pro 6.0 (WaveMetrics) (*Wohland et al., 2001*). The procedure of calibration and quantification of diffusion coefficient and dissociation constants were as previously described (*Foo et al., 2012a*; *Shi et al., 2009a*).

## Gradient analysis

EGFP fusion proteins were excited with a 488 nm laser beam and the emitted fluorescence was collected through a 10X objective lens (Olympus, UPLSAPO NA = 0.40) and a long-pass 505 emission filter with a 2.5X digital zoom. Images were acquired in planes ~ 15 μm below the enveloping layer of the embryos at 512 × 512 pixels with a corresponding size of 1.4 μm²/pixel. Acquired images were analyzed using the ImageJ package. A rectangular region of interest (ROI) with a fixed height of 50.4 μm (36 pixels) adjacent to the source was drawn. The width of the ROI differed depending on the size of the embryo. Windows of 7 × 50.4 μm² (5 × 36 pixels) were binned and the average intensity of each binned window was calculated. Background auto-fluorescence was estimated from images of un-injected embryos and subtracted from all measurements. The data was normalized to the value closest to the source boundary, plotted on the intensity-distance coordinate with ImageJ. The data was pooled and fitted, or individual data sets were fitted and the gradient length was calculated as the mean of all fits. Both procedures yielded similar results. Fits were performed with an exponential decay:

$$C(x) = A * exp\left[-\frac{x}{\lambda}\right] + C \tag{7}$$

where $A$ is the amplitude of the gradient, $\lambda$ is the gradient decay length and $C$ is a possible offset.

## Simulation of the Nodal gradient

Simulations were performed with *Mathematica* 10.0 (Wolfram, Champaign, IL). Initial simulations to determine effective diffusion coefficients in the presence of cells as obstacles (tortuosity) and morphogen binding were conducted in 3D. For this purpose, we simulated a 3D slab of 2 μm height ($z$-axis), 44 μm width ($y$-axis), and 86.7 μm length ($x$-axis), for 5 s (*Figure 5A,B*). We used a diffusion coefficient of $D = 60$ μm$^2$/s, and created 1000 particles at the left border of the simulation volume. The particles were allowed to perform a random walk for 5 s with a time resolution of 5 ms per step. At the left and right borders (along the $x$-axis), particles were reflected. At the other four borders we used periodic boundary conditions. Based upon actual measurements from early gastrula embryos, we assumed that the space is packed with cells of $\sim$10 μm diameter, and an intercellular space (cell membrane-to-cell membrane distance) of $\sim$2 μm. As the height of the simulation volume was only 2 μm, we used cylinders to represent the cells within this space. Under these circumstances, ligand diffusion was reduced by a factor of 1.84. This value is consistent with the findings of Müller et al. who reported tortuosity to reduce diffusion maximally by a factor of 2 (*Müller et al., 2013*). Therefore, for further modeling we assumed the effective diffusion coefficient of the Nodals to be $\sim$30 μm$^2$/s. In the case of binding we used the values in *Table 1* and *Equation 1* to determine the average number of free particles at each step. All concentration profiles were normalized and fitted by the following equation to determine the effective diffusion coefficient:

$$C(x,t) = exp\left[-\frac{x^2}{4D_{eff}t}\right] \tag{8}$$

Here $C(x, t)$ is the concentration profile, $x$ is the coordinate along which the particle diffusion is observed, $t$ is the time at which the profile is measured (i.e., 5 s), and $D_{eff}$ is the effective diffusion coefficient.

Final simulations, including continuing particle production and degradation, used the values given in *Table 1* and were run in 1D with an extent of 200 μm, assuming a reduced diffusion coefficient of 30 μm$^2$/s due to tortuosity, and an average number of particles bound as determined by *Equation 1*.. To ensure that the gradients reached equilibrium, the simulation time was 16 hr 40 min. The normalized concentration gradients $C(x)$ were fitted with a simple exponential function to determine the gradient length $\lambda$.

$$C(x) = exp\left[-\frac{x}{\lambda}\right] \tag{9}$$

## Acknowledgements

We thank members of the Sampath and Wohland laboratories, Rob Cross, Tim Saunders, and Hugh Woodland for discussions and suggestions; Michael Brand for sharing plasmids; Agnieszka Nagorska provided technical assistance; Patrick Müller provided critical comments on the early online version; WY and KS are supported by WMS, WX is the recipient of a NUS graduate Scholarship, KS acknowledges funding from the BBSRC and TW acknowledges funding from the Ministry of Education of Singapore (grant MOE2012-T2-1-101: R-154-000-543-112).

## Additional information

### Funding

| Funder | Grant reference number | Author |
| --- | --- | --- |
| University of Warwick | | Yin Wang |
| National University of Singapore | | Xi Wang |

| Ministry of Education - Singapore | MOE2012-T2-1-101: R-154-000-543-112 | Thorsten Wohland |
| Biotechnology and Biological Sciences Research Council | | Karuna Sampath |

The funders had no role in study design, data collection and interpretation, or the decision to submit the work for publication.

## Author contributions

YW, Experimental design, Zebrafish and cell culture experiments, Manuscript preparation, Acquisition of data, Analysis and interpretation of data; XW, Experimental design, FCS/FCCS measurements, Input for manuscript preparation, Acquisition of data, Analysis and interpretation of data; TW, KS, Experimental design, Manuscript preparation, Acquisition of data, Analysis and interpretation of data

## Author ORCIDs

Karuna Sampath, http://orcid.org/0000-0002-0729-1977

## Ethics

Animal experimentation: This study was performed in strict accordance with institutional animal care regulations and protocols of the National University of Singapore and the University of Warwick (PPL number 70/7836 and PIL number 70/26057 to KS).

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
