## [Decision Letter]

Thank you for submitting your work entitled "Extracellular Interactions and Ligand Degradation Shape the Nodal Morphogen Gradient" for consideration by *eLife*. Your article has been favorably evaluated by K VijayRaghavan (Senior editor) and three reviewers, one of whom, Didier Stainier, is a member of our Board of Reviewing Editors, and another is Hiroshi Hamada.

The reviewers have discussed the reviews with one another and the Reviewing Editor has drafted this decision to help you prepare a revised submission.

Summary:

This paper is concerned with the formation of the Nodal gradient during zebrafish development but has general implications for the formation of morphogen gradients and embryonic patterning. These questions (formation of morphogen gradients and embryonic patterning) are fundamental to our understanding of developmental processes and thus have been under active investigation in the past decades.

This paper breaks new ground in 3 different areas: 1) in vivo measurements of the binding affinity of the Nodal ligands for their major receptor, Acvr2b; 2) measurement of the diffusion coefficient of Nodal ligands and lefty inhibitors in live zebrafish embryos; and 3) examination of the degradation rates of the Nodal ligands. The main, and novel, conclusion is that diffusivity alone is not sufficient to explain the formation of the Nodal gradient, and that the interactions with the receptors and inhibitors, as well as selective ligand destruction are major players in forming the Nodal gradient.

Essential revisions:

The reviewers have agreed on the following additional work: modeling work to test the validity of their measurements/model or additional biophysical measurements (e.g., FRAP) (the latter just in case this is something you already started doing), as outlined in reviewer 2's comments.

Please also make sure to incorporate within the manuscript a discussion of the caveats associated with over-expressing the ligands ectopically (we understand that this is how all the related studies have been carried out in zebrafish but of course, these caveats should be clearly stated and the conclusions appropriately tempered.)

All additional concerns should be addressable by modifying the text.

*Reviewer #1:*

This paper is concerned with the formation of the Nodal gradient during zebrafish development but has general implications for the formation of morphogen gradients and embryonic patterning. These questions (formation of morphogen gradients and embryonic patterning) are fundamental to our understanding of developmental processes and thus have been under active investigation in the past decades.

This paper breaks new ground in 3 different areas: 1) in vivo measurements of the binding affinity of the Nodal ligands for their major receptor, Acvr2b; 2) measurement of the diffusion coefficient of Nodal ligands and lefty inhibitors in live zebrafish embryos; and 3) examination of the degradation rates of the Nodal ligands. The main, and novel, conclusion is that diffusivity alone is not sufficient to explain the formation of the Nodal gradient, and that the interactions with the receptors and inhibitors, as well as selective ligand destruction are major players in forming the Nodal gradient. While some of these conclusions may seem logical/intuitive, the underlying data appear to be solid and the role of ligand degradation surprising.

Additional effort should be placed to write the paper for a broader audience (e.g., add a sentence in figure legend to explain the injection procedure; explain acronyms in figure legend (e.g., ACF, CCF); show structure of sec-EGFP in Figure 1 where it is first used; etc.).

Recent studies on Nodal signaling in zebrafish have mostly used ectopic expression in the animal pole, which is not the endogenous source of Nodal expression. One should probably temper the conclusions from these studies, especially as cellular properties such as the degradation machinery might be different in the poles compared to the margin. Similarly, the stability data generated in this paper were obtained using HEK293T cells, which may be quite different from zebrafish embryonic cells.

*Reviewer #2:*

Wang et al. use FCS and FCCS in zebrafish embryos to measure the diffusion of Nodal ligands and their binding affinity to transmembrane receptors and Lefty. The authors conclude that degradation and Nodal-receptor/inhibitor interactions are important for shaping the Nodal gradient and that the measured diffusion coefficient cannot explain the gradient shape.

The strength and novelty in this work is the in vivo measurements of binding affinity of Nodals to their receptors and antagonists and the new measurements of in vivo diffusivity of Nodal ligands. Nevertheless, I think there are caveats in the strategy the authors have adopted and there are several weaknesses in the data. Overall my opinion is that the study is somewhat preliminary and descriptive. To be of broader interest the biophysical measurements made in this study need to be incorporated into a coherent, self-consistent model for Nodal spread that tests the main conclusions the authors make.

In general I am skeptical about the sole use of FCS for assessing morphogen diffusion properties. FCS measurements rely on very small length and time scales. In this context it is not surprising that measurements of proteins in extracellular space produce diffusion coefficients consistent with the molecular size of the protein measured. To be useful, FCS measurements need to be coupled with measurements at long time/length scales (e.g. FRAP etc.) in order to understand the diffusivity properties of the ligand at scales relevant to tissue patterning.

Previous work from the Schier lab (Müller et al. 2012) have made measurements of some of the same parameters as Wang et al. Wang et al. mention these, but it is difficult to understand from the discussion the similarities and possible reasons for the discrepancies between the two sets of data.

A limitation of the approach the authors take is that it relies on ectopic expression of the proteins assayed. This could introduce various artefacts by taking measurements of proteins expressed from ectopic cell types or located in abnormal regions of the embryo. For example, if ligand secretion/modification machinery are missing or different in the injected cells. Similarly, the measured degradation rates might be affected by over expression, etc.

With the measurements the authors have made, they could attempt to fit and test a biophysical model of Nodal spread that takes account of extracellular movement, clearance and receptor and antagonist binding. This exercise would offer a consistency check that the various measurements are compatible with each other and don't predict physically unreasonable behaviours. It might also produce testable predictions – for example suggest what proportion of ligand would be expected to be receptor bound/free at any instant in time.

*Reviewer #3:*

Among nodal homologs in zebrafish, Sqt acts over a long distance while Cyc functions within in a short distance. However, the mechanism that determines the difference in their signaling range remains unclear. The authors have addressed this issue and have examined diffusivity, binding affinity to their receptor, and stability (degradation rate) of Sqt and Cyc in comparison. Their results show that the diffusivity is similar whereas degradation rate and affinity to the receptor are different. The authors suggest that the binding affinity to the receptor and selective ligand degradation determines the signaling range.

The data presented in this paper look convincing and support their conclusion. It would be much nicer if one can examine their stability in embryo (instead of a culture cell system), but this is probably a realistic way to address this. Although this reviewer is unable to fully evaluate the data obtained by fluorescence correlation spectroscopy and fluorescence cross-correlation spectroscopy for technical reason, I support this paper in principle.

[Editors' note: further revisions were requested prior to acceptance, as described below.]

Thank you for resubmitting your work entitled "Extracellular Interactions and Ligand Degradation Shape the Nodal Morphogen Gradient" for further consideration at *eLife*. Your revised article has been favorably evaluated by K VijayRaghavan (Senior editor), a Reviewing editor, and Reviewer #2. The manuscript has been improved but there are some remaining issues that need to be addressed before acceptance, as outlined below:

As you will see below, Reviewer #2 is concerned about the receptor concentration used for the simulations, especially in light of previous papers (although of unrelated proteins) such as:

Fujioka A, Terai K, Itoh RE, Aoki K, Nakamura T, Kuroda S, Nishida E, Matsuda M., Dynamics of the Ras/ERK MAPK cascade as monitored by fluorescent probes. J Biol Chem. 2006 Mar 31 281(13):8917-26.

Lee E, Salic A, Krüger R, Heinrich R, Kirschner MW. The roles of APC and Axin derived from experimental and theoretical analysis of the Wnt pathway. PLoS Biol. 2003 Oct;1(1):E10.

*Reviewer #2 (General assessment and major comments (Required)):*

The additions and changes that Wang et al. have made to their study address many of the issues raised in the initial review and substantially strengthen the study.

However, these changes do lead to several new questions:

The simulations (Figure 5) that the authors have added greatly enhance the paper. The parameterization of these rely on several assumptions, in addition to the measurements. On the whole these seem well justified. However, whether the assumption of a receptor concentration of 40microM is appropriate is unclear. It would be useful to express this as molecules per cell and to provide citations to any supporting evidence for this concentration. It would also be useful to explain the consequence for the simulations of order of magnitude differences in receptor concentration.

If I understand the simulations correctly, they predict that it would take a considerable time for the gradient to reach steady state. It would be interesting to know how long it takes for the simulations to reach ~80% and ~95% of their steady state. If the time to steady state is of the order (or longer) than mesoderm induction (1-2h) then highlighting this and discussing the consequences in the Discussion will be interesting to the field.

The authors provide a clear explanation in their rebuttal to referee 1 of the differences in use of FCS and FRAP in measuring morphogen gradients. Incorporating elements of this into the Discussion as a way to explain the caveats of the approach would strengthen the discussion.

---

## [Author Response]

Essential revisions:

The reviewers have agreed on the following additional work: modeling work to test the validity of their measurements/model or additional biophysical measurements (e.g., FRAP) (the latter just in case this is something you already started doing), as outlined in reviewer 2's comments.

Please also make sure to incorporate within the manuscript a discussion of the caveats associated with over-expressing the ligands ectopically (we understand that this is how all the related studies have been carried out in zebrafish but of course, these caveats should be clearly stated and the conclusions appropriately tempered.).

All additional concerns should be addressable by modifying the text.

We are pleased that the reviewers found our manuscript on the Nodal morphogen gradient in zebrafish to have general implications for formation for morphogen gradients, to be fundamental to our understanding of developmental processes, and that it breaks new ground in 3 different areas.

In the revised Wang et al. manuscript, we have fully addressed the reviewers’ comments without adding significantly to the manuscript length:

1) We have performed mathematical modeling to test the validity of our measurements, and the models proposed to generate the Nodal morphogen gradient.

2) We show that our measurements are consistent with the computational simulations, and fit with reported values.

3) The revised manuscript has been rewritten for a broad audience, and

4) The revised conclusions take into consideration the mammalian cell and zebrafish ectopic expression assays used.

We show simulations from mathematical modeling in new Figure 5 and new Video 1, Video 2 and Video 3, and the simulation parameters are listed in new Table 1.

We address the specific comments from the individual reviewers below.

Reviewer #1:

*This paper is concerned with the formation of the Nodal gradient during zebrafish development but has general implications for the formation of morphogen gradients and embryonic patterning. These questions (formation of morphogen gradients and embryonic patterning) are fundamental to our understanding of developmental processes and thus have been under active investigation in the past decades.*

*This paper breaks new ground in 3 different areas: 1) in vivo measurements of the binding affinity of the Nodal ligands for their major receptor, Acvr2b; 2) measurement of the diffusion coefficient of Nodal ligands and lefty inhibitors in live zebrafish embryos; and 3) examination of the degradation rates of the Nodal ligands. The main, and novel, conclusion is that diffusivity alone is not sufficient to explain the formation of the Nodal gradient, and that the interactions with the receptors and inhibitors, as well as selective ligand destruction are major players in forming the Nodal gradient. While some of these conclusions may seem logical/intuitive, the underlying data appear to be solid and the role of ligand degradation surprising.*

We are pleased that this reviewer found our manuscript on the Nodal morphogen gradient in zebrafish to have general implications for formation for morphogen gradients, to be fundamental to our understanding of developmental processes, and that it breaks new ground in 3 different areas. We have revised the manuscript taking the reviewer’s comments into consideration.

Additional effort should be placed to write the paper for a broader audience (e.g., add a sentence in figure legend to explain the injection procedure; explain acronyms in figure legend (e.g., ACF, CCF); show structure of sec-EGFP in Figure 1 where it is first used; etc.).

This has been done. We have revised Figure 1 to show the structure of sec-GFP, and explain the terms used (see revised Figure 1, revised figure legends and revised main manuscript).The manuscript has been rewritten to address a broad audience (see revised Wang et al. manuscript). We have provided information regarding the techniques used (FCS, SW-FCCS etc.).

Recent studies on Nodal signaling in zebrafish have mostly used ectopic expression in the animal pole, which is not the endogenous source of Nodal expression. One should probably temper the conclusions from these studies, especially as cellular properties such as the degradation machinery might be different in the poles compared to the margin. Similarly, the stability data generated in this paper were obtained using HEK293T cells, which may be quite different from zebrafish embryonic cells.

This has been done. We have revised the Results and Discussion sections to take into consideration the ectopic assays used (see subsection “Simulation of the Nodal gradient” and Discussion, fourth and sixth paragraphs). We also discuss these in the response to reviewers 2 and 3 comments below.

Reviewer #2:

*Wang et al. use FCS and FCCS in zebrafish embryos to measure the diffusion of Nodal ligands and their binding affinity to transmembrane receptors and Lefty. The authors conclude that degradation and Nodal-receptor/inhibitor interactions are important for shaping the Nodal gradient and that the measured diffusion coefficient cannot explain the gradient shape.*

The strength and novelty in this work is the in vivo measurements of binding affinity of Nodals to their receptors and antagonists and the new measurements of in vivo diffusivity of Nodal ligands. Nevertheless, I think there are caveats in the strategy the authors have adopted and there are several weaknesses in the data. Overall my opinion is that the study is somewhat preliminary and descriptive. To be of broader interest the biophysical measurements made in this study need to be incorporated into a coherent, self-consistent model for Nodal spread that tests the main conclusions the authors make.

We have now amended the manuscript according to the reviewer’s suggestions,and performed simulations of the data (see revised Methods, New Figure 5, new Table 1, and new Video 1, Video 2 and Video 3).

In our study, first we measured in vivo the local diffusivity of the Nodals and their inhibitors Lefty by FCS in early zebrafish embryos. Importantly, our findings resolve a conundrum in light of a recent study by van Boxtel et al., (2015) who reported that contrary to the findings of Müller et al., ligand diffusivity is not a major determinant of the Nodal signals, and that instead, miRNA-430 generates a temporal activation window which is somehow converted into a spatial Nodal activity gradient in zebrafish embryos (van Boxtel et al., 2015).

Our FCS data largely concur with the findings of Müller et al., and show that the Nodals and Lefty have similar local diffusivity *in vivo* in zebrafish embryos.

In addition, we have determined the binding affinity of the Nodal ligands to their major cell surface receptor Acvr2b. Surprisingly, we found that the longer range Nodal, Sqt, binds with higher affinity to the Acvr2 receptor than the short range Nodal, Cyc. This suggests that other factors must determine the effective range of Nodal signals. We then measured the binding affinity of the Nodal factors to the inhibitor Lefty. We also show a role for selective ligand degradation in shaping the Nodal gradient. Finally, we tested the validity of our measurements by computational simulations.

Interestingly, our simulations predict a Sqt gradient of 30 μm and Cyc gradient of 20 μm, which are consistent with theoretical models (Equation 2) and with measured in vivo values (New Figure 5, and Video 1, Video 2, Video 3).Our findings largely support the ‘hindered diffusion’ model proposed by Müller et al. In addition, we found a role for ligand degradation in shaping the Nodal morphogen gradient.

Therefore, our *in vivo* measurements, together with mathematical modeling, and simulations, represent an advance.

In general I am skeptical about the sole use of FCS for assessing morphogen diffusion properties. FCS measurements rely on very small length and time scales. In this context it is not surprising that measurements of proteins in extracellular space produce diffusion coefficients consistent with the molecular size of the protein measured. To be useful, FCS measurements need to be coupled with measurements at long time/length scales (e.g. FRAP etc.) in order to understand the diffusivity properties of the ligand at scales relevant to tissue patterning.

Previous work from the Schier lab (Müller et al. 2012) have made measurements of some of the same parameters as Wang et al. Wang et al. mention these, but it is difficult to understand from the discussion the similarities and possible reasons for the discrepancies between the two sets of data.

Müller et al., reported diffusion of Nodal and Lefty by FRAP (2012) and by FCS (2013). Importantly, our FCS measurements are in the same order of magnitude, and our conclusions largely concur with Müller et al., 2013 in that the local diffusivity of the Nodals and Lefty is similar, although the absolute diffusion coefficient values reported by the two groups differ slightly {e.g., ~40 +/- 11 μm^2^/s (n=14) for Sqt by Müller, compared to our measurements of ~64 +/- 14 μm^2^/s (n=29)}.

The “hindered diffusion” model by Müller et al., 2013 proposes that mobility of the Nodals is slowed down by transient binding interactions of the ligands to unknown diffusion regulators in the extracellular space.

In our study, we measured the diffusion coefficient of the Nodals and Lefty by FCS. In addition we have determined the affinity of the Nodals to the cell surface receptor Acvr2 and Lefty inhibitor by FCCS. We also show by mathematical modeling that diffusion of the Nodals is indeed retarded upon binding, as predicted for hindered diffusion.

Our simulations more or less concur with the predictions of Müller (2013): we find a 1.84-fold reduction in mobility with cells as obstacles (Video 1, Video 2 and new Figure 5 of revised Wang et al.) compared to a presumed maximum 2-fold decrease owing to “tortuosity” (Müller et al. 2013). We then show that mobility/diffusion is reduced further when we take binding into consideration (Video 3).

Overall, our FCS and FCCS measurements reporting binding affinities of the Nodals to the receptor and inhibitor, together with our modeling and simulations largely support the “hindered diffusion by tortuosity and transient binding”model proposed by Müller et al.

A key difference between our findings and model, and that of Schier and colleagues is that they presumed that the clearance/degradation of the molecules does not significantly affect the Nodal gradient (Müller et al., 2012), which is not what we find. We identified a lysosomal targeting region in Cyc that renders this Nodal unstable compared to Sqt (Tian et al., 2008). We had previously shown that this region is responsive to chloroquine, which inhibits lysosomes. The lysosome-targeting region of Cyc when introduced into Sqt, reduces the activity of chimeric Sqt-Cyc fusion proteins. We also find that the slope of the gradient changes with differential ligand stability, with Sqt-Cyc fusions showing a steeper gradient than Sqt (Figure 4;λ of 30 μm for Sqt versus 23 μm for Sqt^Cyc2^). This is supported by assays for signaling range in embryos (Figure 1and Figure 1—figure supplement 1).

Consistent with our results, in our simulations, we find that the exact ligand amount does not change the outcome significantly for concentration changes within a factor of 5 (see revised Results and Discussion). The differences in the gradient length are therefore, a result of differential binding affinities of Sqt and Cyc, and different degradation rates. Thus, degradation of the ligands is a key regulator of the Nodal gradient.

Taken together, our findings show that hindered diffusion via extracellular binding of the Nodal ligands to the Acvr2 receptor and Lefty inhibitor, together with selective ligand degradation play important roles in shaping the Nodal morphogen gradient. We have now amended the Results, Discussion and conclusions to make this clear.

A limitation of the approach the authors take is that it relies on ectopic expression of the proteins assayed. This could introduce various artefacts by taking measurements of proteins expressed from ectopic cell types or located in abnormal regions of the embryo. For example, if ligand secretion/modification machinery are missing or different in the injected cells. Similarly, the measured degradation rates might be affected by over expression, etc.

We agree that ideally one should assay proteins produced by the endogenous loci, at endogenous levels, at native locations in the embryo. Using new CRISPR/Cas based homologous recombination (Hoshijima et al., 2016) to generate single copy reporters to tag endogenous proteins, together with new/emerging 3D imaging techniques (e.g., SPiM) might provide more robust measurements, and indeed such experiments are under way.

In the current study, for ease of analysis, as is commonly performed in zebrafish embryos, we used tagged fusions of ligand and inhibitors that were expressed ectopically in the animal pole, where any interference from other molecules could potentially be minimized. A caveat with such experiments is that the ectopic locations might not contain all the necessary components, or might introduce some artifacts.

However, our current and previous analysis largely concurs with previous work which reported that Cyc acts at short range: Genetic analysis using transplantations of mutant or wild type cells suggested that the range of the Cyc signal is short range, i.e., 1-2 cells from its source (Hatta et al., 1991; Sampath et al., 1998). Chen and Schier found that the range of signaling by Cyc is much less than that of Sqt (2001). We reproduce this in our ectopic animal pole expression assays (see Figure 1 and Figure 1—figure supplement 1). Similarly, our analysis of the gradient also show that compared to the long range Sqt, chimeric Sqt^Cyc2^ protein has a steeper gradient which is consistent with its reduced signaling range (Figure 4 and Figure 1—figure supplement 1). Our modeling of the data measured broadly agrees with the reported in vivo range for the proteins (Figure 5, revised Results and revised Discussion). We also infer that some additional binding events (e.g. to the co-receptor or other molecules) might further influence the gradient.

Thus, our measurements in the animal pole (although not ideal) have provided very useful parameters.

With the measurements the authors have made, they could attempt to fit and test a biophysical model of Nodal spread that takes account of extracellular movement, clearance and receptor and antagonist binding. This exercise would offer a consistency check that the various measurements are compatible with each other and don't predict physically unreasonable behaviours. It might also produce testable predictions – for example suggest what proportion of ligand would be expected to be receptor bound/free at any instant in time.

This has been done. We performed simulations of the data (see new Figure 5, new Table 1, Video 1, Video 2, Video 3). We have inferred the amount of free versus bound ligand from our FCS and FCCS measurements and provide this data in new Table 1 caveat of our model is that we have not considered how interactions of Nodals with Lefty inhibitors impacts in vivo signaling output. We also have not taken account of potential Nodal-Oep/Cripto interactions. Nonetheless, our measurements are largely supported by the theoretical predictions, mathematical simulations, and fit well with reported values.

Reviewer #3:

Among nodal homologs in zebrafish, Sqt acts over a long distance while Cyc functions within in a short distance. However, the mechanism that determines the difference in their signaling range remains unclear. The authors have addressed this issue and have examined diffusivity, binding affinity to their receptor, and stability (degradation rate) of Sqt and Cyc in comparison. Their results show that the diffusivity is similar whereas degradation rate and affinity to the receptor are different. The authors suggest that the binding affinity to the receptor and selective ligand degradation determines the signaling range.

The data presented in this paper look convincing and support their conclusion. It would be much nicer if one can examine their stability in embryo (instead of a culture cell system), but this is probably a realistic way to address this. Although this reviewer is unable to fully evaluate the data obtained by fluorescence correlation spectroscopy and fluorescence cross-correlation spectroscopy for technical reason, I support this paper in principle.

We thank the reviewer for the supportive comments. Owing to the labile nature of Cyc, for ease of assessing degradation, cell culture was performed. Nonetheless, we recognize the reviewers concerns regarding measurements of stability in cell culture rather than in embryos, and therefore, in our mathematical simulations, we now use previously reported half-life values for Cyc and Sqt (see new Table 1).

For all FCS and FCCS measurements, we used appropriate controls (e.g., sec-eGFP and eGFP-mCherry fusions), and all measurements were first tested using reagents previously shown to determine diffusion of FGF in zebrafish embryos (Yu et al., 2009).

[Editors' note: further revisions were requested prior to acceptance, as described below.]

The manuscript has been improved but there are some remaining issues that need to be addressed before acceptance, as outlined below:

As you will see below, Reviewer #2 is concerned about the receptor concentration used for the simulations, especially in light of previous papers (although of unrelated proteins) such as:

Fujioka A, Terai K, Itoh RE, Aoki K, Nakamura T, Kuroda S, Nishida E, Matsuda M., Dynamics of the Ras/ERK MAPK cascade as monitored by fluorescent probes. J Biol Chem. 2006 Mar 31 281(13):8917-26.

Lee E, Salic A, Krüger R, Heinrich R, Kirschner MW. The roles of APC and Axin derived from experimental and theoretical analysis of the Wnt pathway. PLoS Biol. 2003 Oct;1(1):E10.

Reviewer #2 (General assessment and major comments (Required)):

*The additions and changes that Wang et al. have made to their study address many of the issues raised in the initial review and substantially strengthen the study.*

However, these changes do lead to several new questions:

The simulations (Figure 5) that the authors have added greatly enhance the paper. The parameterization of these rely on several assumptions, in addition to the measurements. On the whole these seem well justified. However, whether the assumption of a receptor concentration of 40microM is appropriate is unclear. It would be useful to express this as molecules per cell and to provide citations to any supporting evidence for this concentration. It would also be useful to explain the consequence for the simulations of order of magnitude differences in receptor concentration.

We are pleased that the reviewer found the revised Wang et al. manuscript improved and that the simulations greatly enhance the paper.

We determined from our simulations that a receptor concentration of 40 μM is required to slow down diffusion sufficiently to obtain a 30 μm Sqt gradient. However, it is difficult at this time to deduce a single receptor number per cell for several reasons:

i) First, this number depends on the ratio of interstitial space volume to cell size and number. So any change in the estimated distance between cells (e.g., by cell division, movement and growth) would alter that number. However, the effective diffusion coefficient (D_eff_) does not depend on this; D_eff_ depends only on the concentration and tortuosity, and we have used this in our estimates.

ii) Second, at the moment we do not know whether there are additional binding sites in the interstitial space as reported for other morphogens (e.g. extracellular HSPG binding by Fgf8; Yu et al., 2009). The binding constant and concentration of such sites, or amendments to the binding constants of the morphogens to receptors would lead to changes in that number.

iii) Third, the degradation/clearance rate is the main determinant of this number and has a direct influence on the receptor concentration: a faster clearance would lead to faster establishment of the gradient and also reduce the receptor number required to establish the λ of 30 μm and 20 μm that we observed for Sqt and Cyc. Therefore, we estimate that there could be anywhere between 100,000 to millions of receptors per cell. For now, we prefer to leave this determination to future studies when precise production and clearance rates for the various molecules become available, and when it is known if there are additional binders for the ligands.

We have now amended the Results section to state the following:

“The estimation of the receptor number of 40 μM is based on the value of D_eff_ required to establish the gradient of appropriate dimensions for Sqt. […] Based on the above, we assume that 40 μM is the upper limit for the receptor concentration”.

If I understand the simulations correctly, they predict that it would take a considerable time for the gradient to reach steady state. It would be interesting to know how long it takes for the simulations to reach ~80% and ~95% of their steady state. If the time to steady state is of the order (or longer) than mesoderm induction (1-2h) then highlighting this and discussing the consequences in the Discussion will be interesting to the field.

The time to reach the steady state is determined by the clearance rate, k: (1-Exp(-k*t)). Therefore, for Cyc, 80% and 95% of steady state levels are achieved at 0.7 and 1.25 h, respectively, which is consistent with the timing for mesoderm induction in the gastrula. By the same predictions, Sqt reaches the 80% and 95% levels at ~4 and 7 h, respectively, which is longer than expected. We should add that we have not taken into consideration cell divisions etc. during that time or binding to other factors, and how such factors might influence the gradient.

We have now incorporated this into the revised Discussion (fifth paragraph).

The authors provide a clear explanation in their rebuttal to referee 1 of the differences in use of FCS and FRAP in measuring morphogen gradients. Incorporating elements of this into the Discussion as a way to explain the caveats of the approach would strengthen the discussion.

This has been done (Discussion section).